# Multiomics Analysis of a *DNAH5*-Mutated PCD Organoid Model Revealed the Key Role of the TGF-β/BMP and Notch Pathways in Epithelial Differentiation and the Immune Response in *DNAH5*-Mutated Patients

**DOI:** 10.3390/cells11244013

**Published:** 2022-12-12

**Authors:** Wenhao Yang, Lina Chen, Juncen Guo, Fang Shi, Qingxin Yang, Liang Xie, Danli Lu, Yingna Li, Jiaxin Luo, Li Wang, Li Qiu, Ting Chen, Yan Li, Rui Zhang, Lu Chen, Wenming Xu, Hanmin Liu

**Affiliations:** 1Department of Paediatric Pulmonology and Immunology, West China Second University Hospital, Sichuan University, Chengdu 610000, China; 2Key Laboratory of Birth Defects and Related Diseases of Women and Children (Sichuan University), Ministry of Education, Chengdu 610000, China; 3NHC Key Laboratory of Chronobiology (Sichuan University), Chengdu 610000, China; 4The Joint Laboratory for Lung Development and Related Diseases of West China Second University Hospital, Sichuan University and School of Life Sciences of Fudan University, West China Institute of Women and Children’s Health, West China Second University Hospital, Sichuan University, Chengdu 610000, China; 5Sichuan Birth Defects Clinical Research Centre, West China Second University Hospital, Sichuan University, Chengdu 610000, China; 6Department of Obstetrics/Gynaecology, Joint Laboratory of Reproductive Medicine (SCU-CUHK), Key Laboratory of Obstetric, Gynaecologic, and Paediatric Diseases and Birth Defects of Ministry of Education, West China Second University Hospital, Sichuan University, Chengdu 610000, China

**Keywords:** airway organoid, primary ciliary dyskinesia, immune response, DNAH5, multiomics analysis

## Abstract

Dynein axonemal heavy chain 5 (DNAH5) is the most mutated gene in primary ciliary dyskinesia (PCD), leading to abnormal cilia ultrastructure and function. Few studies have revealed the genetic characteristics and pathogenetic mechanisms of PCD caused by DNAH5 mutation. Here, we established a child PCD airway organoid directly from the bronchoscopic biopsy of a patient with the DNAH5 mutation. The motile cilia in the organoid were observed and could be stably maintained for an extended time. We further found abnormal ciliary function and a decreased immune response caused by the DNAH5 mutation through single-cell RNA sequencing (scRNA-Seq) and proteomic analyses. Additionally, the directed induction of the ciliated cells, regulated by TGF-β/BMP and the Notch pathway, also increased the expression of inflammatory cytokines. Taken together, these results demonstrated that the combination of multiomics analysis and organoid modelling could reveal the close connection between the immune response and the DNAH5 gene.

## 1. Introduction

Primary ciliary dyskinesia (PCD) is a systemic disease involving ciliary structural and functional disorders caused by gene defects, with an estimated prevalence of 1:10,000 to 1:15,000 in Europeans but a much higher prevalence in the British Asian population [1,2,3]. The surface of the airway epithelium is distributed with motile “9 + 2” cilia (nine peripheral microtubule doublets with a central pair). The axonemal dynein consists of interdyneatin arms (IDAs), which regulate the waveform, and outer dynein arms (ODAs), which generate the majority of the beating force [4]. The ODAs contain at least two types: type 1 (DNAH11 and DNAH5, located in the proximal part of the axoneme), and type 2 (DNAH9 and DNAH5, distal to the axoneme) [5,6,7]. With rapid advances in our understanding of PCD genetics, more than 40 genes have been confirmed to be involved in the pathogenesis of PCD, and approximately 70% of patients can be effectively diagnosed by genetic testing [8]. Dynein axonemal heavy chain 5 (DNAH5), located along the ciliary axoneme, is the most frequently reported mutated gene in most of the clinical studies [9,10,11], and this gene mutation is also a common cause of primary ciliary dyskinesia with ODA defects, representing the most frequent cause of PCD and the randomization of left–right body asymmetry [12,13]. Although different PCD patients have different gene mutations and molecular pathogenetic mechanisms, they share common clinical features, such as frequent respiratory infections with excessive neutrophil influx. The emerging studies have begun to demonstrate that abnormal neutrophil activity and increased IL-1β, TNF-α and IL-8 are produced by monocytes in patients with PCD [14]. These results suggest that there are exacerbated inflammatory reactions in PCD patients compared with healthy individuals [15,16]. However, the indirect and direct correlation between specific ciliated gene deficiency and an abnormal immune system remains elusive.

To reveal the molecular mechanism of PCD, gene-knockout mice, zebrafish and mouse airway organoids have been used to mimic the characteristics of PCD with typical mutations [11,12,16]. The long time and high costs of genetic models have limited basic research on pathogenetic mechanisms. Meanwhile, although these animal models could mimic some characteristics of PCD, human PCD features are difficult to fully recapitulate, due to species differences and the complexity of PCD phenotypes. There is an urgent need to develop humanized and patient-specific models to better mimic human PCD aetiology and understand the pathophysiological mechanism. To date, PCD patient-derived epithelial cells have been cultured in the 2D culture method or air–liquid interface (ALI) culture system to maintain specific phenotypes and auxiliary diagnosis [17,18]. Primary epithelial cells could maintain the ability to proliferate and differentiate in vitro for a greater number of passages. Several key characteristics, including mucociliary differentiation, gene deficiency and specific receptors for virus infection, could be retained, making it suitable for airway disease- and viral infection-related studies [19,20,21]. However, after long-term expansion and cryopreservation, these cells gradually lose their initial features. The air–liquid interface culture (ALI) system is another method to support pseudostratified airway epithelium, mainly including ciliated and goblet cells. ALI cells cannot be passaged, and the differentiation process is easily affected by the culture conditions. Nevertheless, the amount of human tissue obtained from patient biopsies greatly limits the in vitro studies relying on these culture methods.

Organoids are self-organizing 3D structures grown from tissues from patient biopsies and can better reflect organ structure and function than traditional cell culture. Furthermore, patient-specific organoids remain genetically and phenotypically stable for an extended time. Recent studies have shown that mouse and human airway organoids could be used to model PCD-related genetic disease [16,22] and have a wide range of applications, from basic research and drug discovery to disease diagnosis.

In this study, we identified a typical PCD paediatric patient with a *DNAH5* gene mutation and established DNAH5-mutated PCD airway organoids derived from tracheobronchial biopsies. This DNAH5-mutated child airway organoid recapitulated the typical characteristics of PCD airway epithelium in vitro. To reveal the genomic and proteomic molecular features, a combination of single-cell RNA-Seq and proteome analysis from the DNAH5-mutated and normal child airway organoids was performed. Furthermore, these results demonstrate that the immune response was decreased in the DNAH5-mutated organoids compared to the normal organoids. Overall, we evaluated the biological characteristics of patients and airway organoids with DNAH5 mutations.

## 2. Materials and Methods

### 2.1. Patient Information and Informed Consent

The Declaration of Helsinki principles were followed. The study protocol was approved by the Ethical Review Board of West China Second University Hospital, Sichuan University (KL118), and written informed consent was obtained from all the family members involved in this study. The diagnostic criteria for PCD were in accordance with the European Respiratory Society guidelines for the diagnosis of PCD [23].

### 2.2. Human Tissue Processing and Airway Organoid Culture

Child airway tissues, derived from the suspected patient, were collected by bronchoscopy. The residual sample was preserved in DMEM and transported at 4 °C. Upon receiving the sample in the laboratory, the tissues were washed with cold DPBS twice and transferred into a 10-mL digestion medium containing 400 U/mL collagenase I (Sigma, St. Louis, MO, USA, 9001-12-1), 0.25 mg/ml Protease E (Sigma, P5147), 10 μM Y27632 (Selleck, Houston, TX, USA, S6390), and 10 U/ml DNAse I (Sigma, 10104159001) in AdDF+++ buffer (Advanced DMEM/F12 containing 1× Glutamax, 10 mM HEPES, and antibiotics) for 1 h on a shaker at 37 °C. After the enzymatic digestion, the cell suspension was sheared with 1-mL pipettes, filtered with a 40-μm strainer, and centrifuged at 200× *g* for 3 min; the supernatant was discarded. Then, the cell pellet was washed twice with AdDF++ and embedded in Matrigel (Corning, NYC, New York, NY, USA, 356231). A single well of a 24-well plate was seeded with the suspension containing 30 μL of Matrigel and 5000 cells. After solidifying at 37 °C for 15 min, 500 μL of previously published airway organoid culture medium was added to each well, and the plates were incubated under standard culture conditions (37 °C, 5% CO_2_). The child airway organoid culture medium consisted of AdDF+++, 1× B27 (Gibco, Grand Island, New York, NY, USA, 0080085SA), 5 mM nicotinamide (Sigma, N0636), 1.25 mM N-acetylcysteine (Sigma, A0737), 500 ng/mL R-spondin1 (R&D, Minneapolis, MN, USA, 4645), 25 ng/mL recombinant human FGF7 (PeproTech, NJ, USA, 450-61), 100 ng/mL recombinant human FGF10, 100 ng/mL recombinant human Noggin (R&D, 6057), 5 μM Y27632 (CST, Boston, MA, USA, 13624), 500 nM SB202190 (Selleck, S1077), and 500 nM A-8301 (Selleck, S8301). The culture medium was changed every 4 days.

The airway organoids were passaged when the diameter of the organoid was 100–200 μm. All the organoids and the Matrigel were resuspended in AdDF+++ buffer. After centrifugation at 200× *g* for 3 min, the organoid pellets were resuspended with 1 mL of TrypLE 1× (Gibco, A1217701), incubated for 5–10 min at 37 °C, and mechanically sheared with 1-mL pipettes. After washing twice with AdDF+++, the cell pellets were filtered with a 40-μm strainer, resuspended in Matrigel and reseeded in another well. The single cells were seeded at a density of 3000–4000 cells per well in a 24-well plate.

### 2.3. RSV Infection

RSV A2 (ATCC, Manassas, VA, USA VR-1540) was obtained from Chongqing Medical University and propagated in HEp-2 cells (ATCC), and the viral titre was determined according to a published protocol [24]. The airway organoids were prepared at a density of 3000 cells per well in a 24-well plate. After washing in cold AdDF+++, a well of organoids was infected with an incubation solution (200 μL of organoid culture medium plus 2 μL of virus with 1 × 10^8^ pfu/mL) on a single 48-well plate. After incubating for 6 h at 37 °C and 5% CO_2_, the infected organoids were washed twice with AdDF+++ and seeded as described above.

### 2.4. RNA Preparation and qRT–PCR

Total RNA was extracted from airway organoids using an RNAprep Pure Micro Kit (Tiangen, Beijing, China, CA, DP420). The cDNA was synthesized using a Transcriptor First Strand cDNA Synthesis Kit (Roche, Basel, Switzerland, 04897030001), according to the manufacturer’s protocol. A Q-PCR was performed by using SYBR green (Promega, Madison, WI, USA, A600A) and Bio-Rad systems. The gene expression was quantified using the ΔΔCt method and normalized to GAPDH. The primers are listed in Appendix A.

### 2.5. Droplet Digital PCR

All the Droplet Digital PCR (ddPCR) procedures followed the manufacturer’s instructions for the QX200 Droplet Digital PCR System using the supermix for probes (no dUTP) (Bio-Rad, Hercules, CA, USA). The final reaction volume was 20 µL and included the following: 10 µL of 2× supermix for probes (no dUTP) (Bio-Rad); 2 µL of cDNA generated by 100 ng of RNA of the target sample; 0.5 µL of RSV-N-F; 0.5 µL of RSV-N-R; 0.5 µL of RSV-N-P; and 6.5 µL of nuclease-free water. Then, the 20-μL mixture was converted to droplets with a QX200 droplet generator (Bio-Rad). Droplet-partitioned samples were then transferred to a 96-well plate, sealed and cycled in a T100 Thermal Cycler (Bio-Rad) under the following cycling protocol: 95 °C for 10 min (DNA polymerase activation); 40 cycles of 94 °C for 30 s (denaturation) and 58 °C for 1 min (annealing); 98 °C for 10 min; and a final hold at 4 °C. The cycled 96-well plate was then transferred and read in the FAM and HEX channels using the QX200 reader (Bio-Rad).

### 2.6. Transmission Electron Microscopy

TEM images of child airway biopsies were obtained from the Department of Pathology, West China Hospital. Sichuan University. The child airway organoids were fixed in 3% glutaraldehyde and processed by the Chengdu Lilai Biomedicine Experiment Center following a conventional protocol. The final cilia of ultrathin tracheal epithelial cell sections were observed by TEM (TECNAI G2 F20).

### 2.7. Section Immunofluorescence (IF), Whole-Mount IF and Live Staining

The airway organoids were harvested and fixed in 4% paraformaldehyde at 4 °C overnight. For the section IF, the fixed organoids were dehydrated, paraffin-embedded and sectioned. For the whole-mount IF, the fixed organoids were permeabilized using 0.5% Triton X-100 and blocked with 5% bovine serum albumin (BSA) for 1 h at room temperature (RT). The organoids were then incubated with primary antibodies, including ace-tubulin (Santa Cruz, Santa Cruz, CA, USA, sc-23950), Krt5 (Abcam, CB, Waltham, MA, USA, ab52635), and DNAH5 (Abcam, ab234826), at 4 °C overnight. The next day, the organoids were washed three times and incubated with Hoechst 33342 (Invitrogen, Life Technologies, Carlsbad, CA, USA, H3570) and the secondary antibodies labelled with Alexa Fluor 488 (Invitrogen, A11001) or Alexa Fluor 594 (Invitrogen, A11012) for 1 h at RT. Finally, the organoids were washed three times and sealed with anti-fluorescence quenching sealing tablets (YEASEN, Shanghai, China, CA, 36307ES08). The images were acquired using a laser-scanning confocal microscope (Olympus, Waltham, MA, USA). The quantification of the expression intensities of the different markers was performed using ImageJ software.

### 2.8. High-Speed Microscopy Analysis of Child Airway Organoids

To record the ciliary beating frequency (CBF), the DNAH5-mutation and normal airway organoids were prepared at room temperature (25 °C) for video microscopy with a 40× objective (Sprinter-HD Optronics). Movies were recorded at 200 fps and analysed blindly by two researchers, as described earlier. The CBF was recorded and analysed using video.

### 2.9. Flow Cytometry

The harvested airway organoids were digested with 1×Triple at 37 °C for 20 min. Single organoid cells were obtained by filtering through 40-μm strainers. Flow cytometry was performed according to the protocol from the eBioscience Foxp3/Transcription Factor Staining Buffer Set (Invitrogen, Cat: #00-5523). Acetyl-α-Tubulin (CST, Boston, MA, USA, #81502) and an isotype control (CST, #2985S) were used. A FACS analysis was performed on a BD Fortessa machine. The data analysis was performed using FlowJo (v10.8.1).

### 2.10. Cytokine Array Measurement

The culture medium of child airway organoids infected by RSV was collected at 48 hpi. After centrifugation at 12,000× *g* and 4 °C for 5 min, the supernatant was harvested and stored at −80 °C. The cytokines in the supernatant were measured by a Quantibody human inflammation array 3 (QAH-INF-3, RayBiotech). The experiment was performed according to the manufacturers’ protocol, and the microarray was scanned and tested by an InnoScan 300 Microarray Scanner.

### 2.11. Western Blot

The airway organoids were lysed in ice-cold RIPA buffer (Beyotime, Shanghai, CA, P0013C) containing an inhibitor cocktail (Bimake, Houston, TX, USA, B14001) with ultrasonic waves and incubated on ice for 40 min. Next, the samples were added to a 5× sodium dodecyl sulfate (SDS) loading buffer and heated at 95 °C. The denatured proteins were separated on 3–8% Tris-acetate gel (Thermo Fisher, Waltham, MA, USA, EA0375BOX) and transferred to a 0.45-µm pore size polyvinylidene difluoride (PVDF) membrane (Millipore) at 40 V overnight. Subsequently, the membranes were blocked in 5% skimmed milk for 1 h at RT and incubated with primary antibodies, GAPDH (adcam, ab181602), DNAH5 (adcam, ab234826) and Krt5 (abcam, ab52635) overnight at 4 °C. The following day, the samples were washed and incubated with a goat anti-mouse IgG secondary antibody-HRP (1:5000, Thermo Fisher, 32230) and goat anti-rabbit IgG secondary antibody-HRP (1:5000, Thermo Fisher, 6120) in 5% skimmed milk at RT for 1 h. An ECL chemical substrate (Millipore) was used to visualize the developed immunoblot.

### 2.12. Proteomic Analysis

The proteomic analysis was subjected to an LC–MS/MS analysis with a standard protocol. In brief, (1) the organoids were washed in PBS twice and then sonicated three times on ice using a high-intensity ultrasonic processor (Scientz) in a lysis buffer (8 M urea, 1% protease inhibitor cocktail). To remove the remaining debris, the supernatant was collected after centrifugation at 12,000× *g* at 4 °C for 10 min. The protein concentration was determined with a BCA kit, according to the manufacturer’s instructions. (2) The protein sample was processed after the trypsin digestion and purification by a C18 SPE column. (3) The peptides were dissolved in 0.5 M TEAB. Each channel of peptide was labelled with its respective TMT reagent (Thermo Scientific). (4) The tryptic peptides were first dissolved in solvent A (0.1% formic acid, 2% acetonitrile/in water). The peptides were separated with gradient solvent B (0.1% formic acid in 90% acetonitrile) on an EASY-nLC 1200 UPLC system (Thermo Fisher Scientific, Waltham, MA, USA). Then, the separated peptides were analysed in a Q ExactiveTM HF-X (Thermo Fisher Scientific) with a nanoelectrospray ion source. The fragments were detected in the Orbitrap at a resolution of 30,000. (5) The raw MS/MS data were processed using the MaxQuant search engine (v.1.6.15.0). The tandem mass spectra were searched against the human Swiss-Prot database (20422 entries) concatenated with the reverse decoy database. (6) The data were analysed using the GO annotation (http://www.ebi.ac.uk/GOA/, accessed on 1 Octorber 2021) and the KEGG Pathway 563 annotation (KEGG online service tool KAAS mapper).

### 2.13. Single-Cell Preparation for Sequencing

The organoids were washed twice with PBS. The organoids were digested in 1× Triple for 30 min, and then washed twice with AdDF+++. The single organoid cells were obtained by filtering through 40-μm strainers. The cells were pelleted by centrifugation at 200× *g* for 5 min. The organoid cells were suspended in AdDF+++ at a density of ~1000 cells/µL, suitable for single-cell sequencing.

### 2.14. Single-Cell Sequencing

The cellular suspensions (~6000 cells) were loaded on a Chromium Single Cell Instrument (10X Genomics, Pleasanton, CA, USA) to generate single-cell GEMs. The sample processing and single-cell RNA-Seq library preparation were performed using Chromium Single Cell 3’ v2 reagents. The sequencing was performed on an Illumina NextSeq500 following the manufacturer’s instructions.

### 2.15. Single-Cell Data Analysis

The Cell Ranger Suite version 7.0.0 was used to perform the sample demultiplexing, barcode processing and single-cell gene UMI (unique molecular index) counting (http://software.10xgenomics.com/single-cell/overview/welcome, accessed on 6 March 2022). The single-cell RNA-Seq raw data were filtered according to the following rules: DNAH5 data (nCount_RNA ≥ 750 and nCount_RNA ≤ 100,000 and nFeature_RNA ≥ 750 and nFeature_RNA ≤ 10,000) and CTRL data (nCount_RNA ≥ 550 and nCount_RNA ≤ 100,000 and nFeature_RNA ≥ 400 & nFeature_RNA ≤ 10,000). The percentage of mitochondria in the DNAH5 data is no more than 11% of the total UMI amount, whereas that of the CTRL data is no more than 19%. The gene expression (in UMI) was scale-normalized by transforming in log2 (UMI+1) for the downstream analysis. The principal component analysis (PCA) was carried out on normalized UMI counts, and the viable expressed genes were defined with at least 10 gene UMI counts in at least three percent of all the cells. The t-SNE plot was generated with the Seurat package. A one-way ANOVA and an F test for multigroup comparison using ArrayStudio (www.omicsoft.com/array-studio/, accessed on 12 March 2022) were performed to identify specifically expressed genes of a cell cluster.

### 2.16. Statistical Analysis

Each experiment was replicated at least three times. All the data are expressed as the mean ± standard deviation of the mean. All the statistical analyses were performed using GraphPad Prism software (version 6.01). A normality test was conducted to examine whether all the statistical data adjusted to a normal distribution. To compare the difference between the two groups, t tests were used. For comparisons of two or more groups, a one-way analysis of variance (ANOVA) followed by Dunnett’s multiple comparisons test were used for statistical comparison. When the data were not normally distributed, the Mann–Whitney–Wilcoxon test was used instead of the t test to compare the differences between the groups. A one-way analysis of variance (ANOVA) was used for statistical comparison to compare three or more groups. The significant differences between the groups are represented by * *p* < 0.05.

## 3. Results

### 3.1. The Clinical Characteristics of a Paediatric Patient with a DNAH5 Mutation

A 12-year-old PCD patient with a DNAH5 mutation was identified by clinical symptoms, lung functional tests and exome sequencing. The exome sequencing results showed that the proband had a compound DNAH5 gene mutation with c.1837G>C and c.13486C>T mutations inherited from his mother and father, respectively (Figure 1A,B). The spirometry data from this patient suggested a moderate obstructive ventilation dysfunction and small airway airflow obstruction (Figure 1D). We next questioned whether DNAH5 mutations result in ciliary ultrastructural abnormalities. To answer this question, the patient’s bronchoscopic biopsy was collected and examined by TEM. The results demonstrated the presence of a “9 + 2” axoneme structure and partial defect of the outer dynein arm, consistent with previous studies (Figure 1C). Nasosinusitis and bronchiectasis, the common clinical features in PCD patients, were confirmed by computed tomography (CT) (Figure 1E). These results indicate that the compound DNAH5 mutations could affect the structure of the outer dynein arm and further lead to the disease manifestation of PCD.

### 3.2. The Child Airway DNAH5-Mutated Organoids Modelled Primary Ciliary Dyskinesia

To establish a patient-specific in vitro model, airway epithelial cells derived from a paediatric patient with DNAH5 mutations were collected by bronchoscopy. The obtained restricted human airway sample was prioritized for examination by TEM to confirm the integrity of the ciliary structure, and the residual sample was processed by enzymatic and mechanical digestion and cultured with the previously published airway organoid culture medium within 6 hours after collection. The established patient-specific PCD airway organoid could be long-term expanded, cryopreserved and thawed repeatedly (Figure 2A). Most of the airway organoids were hollow spheres, and the beating cilia could be observed and captured by high-speed microscopy. The airway organoid with the DNAH5 mutation from the patient had an abnormal morphology and a thicker cavity wall than the normal airway organoid from the child, excluding PCD (Figure 2B). Ace-tubulin staining showed that the cilia numbers were significantly reduced in the patient sample with the DNAH5 mutation (Figure 2C and Appendix A). The Western blot results demonstrated the loss of the DNAH5 protein in the DNAH5-mutated airway organoids compared with normal organoids (Figure 2D). The organoid function was significantly affected by the DNAH5 mutation, without beating cilia, as observed by live staining of SPY555-tubulin and high-speed microscopy (Figure 2E,F, Appendix A). Overall, the DNAH5-mutated, patient-derived airway organoids could recapitulate and maintain the genetic makeup and biological phenotype in vitro for an extended time.

### 3.3. Single-Cell Transcriptional Profiles from PCD Organoids

To further define the effects of DNAH5 mutation on different airway epithelial cells, we conducted single-cell RNA sequencing (scRNA-Seq) of the patient’s airway organoids with the DNAH5 mutation and normal organoids. Six cell populations, including basal cells, cycling basal cells, super basal cells, club cells, goblet cells and multiciliated cells, were identified based on the expression of canonical markers, which was consistent with a previous study [25] (Figure 3A and Appendix A). The GO enrichment analysis of various cell populations between the two groups presented diverse biological functions and distinct gene set enrichment. The basal cells and multiciliated cells displayed a common enrichment score for the immune response (Figure 3B,D and Appendix A) and decreased gene expression of immune responses, such as CXCL1, S100P, NPC2, and TIMP2, mainly regulating the innate immune and biological functions of macrophages (Figure 3C,E). Related immune gene changes were also observed in the club and goblet cells (Appendix A). Meanwhile, the regulation of cell morphogenesis and cell-substrate adhesion, which is necessary for regeneration, was also affected by DNAH5 mutation. These results confirm the relationship between the DNAH5 gene and the immune response on diverse epithelial cells, and that DNAH5 mutation may directly impair the first defensive barrier against pathogen invasion on the airway epithelium.

### 3.4. The Compensatory Response Induced by DNAH5 Mutation

After damage has occurred, the compensatory response of the tissue could restore epithelial homeostasis. Some airway injury models have demonstrated that basal cells are an essential component for maintaining and repairing the airway epithelium. The remaining basal cells in the damaged region could increase their proliferation and differentiation ability, which is mainly regulated by Wnt, TGF-β/BMP and Notch signalling. Compensatory phenomena were observed in the DNAH5-mutated child airway organoids by comparing the relative expression of the Wnt, TGF-β/BMP and Notch signalling in basal cells. The lower expression of the Wnt agonist and higher expression of the TGF-β/BMP inhibitor in the basal cells of the DNAH5-mutated organoids favoured the differentiation of ciliated cells compared to the normal organoids (Figure 3C,F). For the multiciliated cells, the motile cilia length is regulated by the TGF-β pathway [26], and the immune response was also affected by the Wnt pathway (Figure 3G). The lower expression of the Wnt and TGF-β pathways may partially account for the impaired cilia function and immune response in the DNAH5-mutated organoids.

### 3.5. Decreased Immune Response in the DNAH5-Mutant Organoid

To further elucidate the molecular mechanisms caused by DNAH5 mutation at the protein level, we performed proteomic analyses of DNAH5-mutated and normal airway organoids. The GO enrichment analysis revealed that the immune response was significantly decreased in the DNAH5-mutated organoids, while the protein expression related to epithelial cell differentiation was increased (Figure 4A). We further analysed the enriched biological processes for differentially expressed genes between DNAH5-mutated and normal organoids. A series of proteins related to the adaptive and innate immune response were downregulated in the DNAH5-mutated airway organoids, whereas the proteins that regulate epithelial cell differentiation were upregulated (Figure 4B). These results suggest that DNAH5 mutation could attenuate the adaptive and innate immune response. To further explore the immune response to the outside pathogen, both organoids were infected with respiratory syncytial virus (RSV), the most common virus of respiratory tract infections in infants and young children (Figure 4C). Our results showed that the RSV-N gene copies in the airway organoids with the DNAH5 mutation were significantly lower than those in the control at the initial infection stage, while the RSV-N gene copies of both the RSV-infected organoids were greatly increased at 48 hpi. However, at 48 hpi, the RSV-N gene copies in the airway organoids with the DNAH5 mutation were significantly higher than those in the control. These results suggest that the balance between epithelial cell differentiation and the immune response in the organoid structures caused by the DNAH5 mutation may contribute to altered defence against viral entry and inhibiting viral replication at the early phase of infection, while the ability to defend against the RSV was instead decreased at the late infection stage. This phenomenon was consistent with the proteomic results, which showed that the DNAH5 mutation may lead to a decreased immune response, as demonstrated above. Subsequently, we sought to understand the mechanisms underlying the crosstalk between the DNAH5 mutation and the immune response. The cytokine profiles of the culture medium (CM) from the DNAH5 mutation and normal airway organoids that were infected with RSV were measured using a RayBio Human Cytokine Antibody Array. Eight cytokines, G-CSF, GM-CSF, MCSF, IL-6, IL-11, TIMP-1, TIMP-2 and TNFR1, were found to be downregulated in the CM derived from the RSV-infected DNAH5-mutated airway organoids compared to the CM derived from the RSV-infected normal airway organoids, whereas one cytokine, IL-8, was upregulated (Figure 4D). The other cytokines were either not detected or not significantly different (Appendix A). Previous studies have demonstrated that the eight downregulated cytokines are closely associated with the immune response process to an RSV infection for the airway epithelial cells [27]. These results indicate that DNAH5 deficiency leads to reduced inflammatory cytokine release and impairs the immune response to RSV infection. Overall, DNAH5 mutation could decrease the immune response ability of the airway epithelium, which may account for the frequent respiratory infections in children diagnosed with PCD.

### 3.6. Establishment of Optimized Cilium-Induced Conditions upon Modulation of the TGF-β/BMP and Notch Pathways

In the airway epithelium, previous studies have demonstrated that cilia differentiation is closely associated with TGF-β/BMP and Notch signal changes, and our single-cell sequencing further showed significant changes in the TGF-β/BMP and Notch pathways in the DNAH5-mutated organoids [22,28,29]. Furthermore, a decrease in the immune response was also observed in the child airway organoids with the DNAH5 mutation at the mRNA and protein levels. However, it is unclear whether the CBF and the immune response are influenced by the TGF-β/BMP and Notch signalling in vitro. To clarify this question and optimize the differentiation medium, we attempted to inhibit the Notch signal and activate the BMP signal separately and in combination. The normal airway organoids were digested into single cells and passaged. After the single cells grew into a hollow sphere for approximately 8 days, the cilia-induced medium was added at D0, and the airway organoids were analysed and harvested at D8 and D16 (Figure 5A). Treatment with DAPT, a gamma-secretase inhibitor, significantly induced FOXJ1 mRNA expression and increased the numbers of ciliated cells in the airway organoids, whereas Scgb1A1 mRNA expression was decreased compared with the control group or other treatments (Figure 5E,F). Meanwhile, the FOXJ1 mRNA expression and the numbers of ciliated cells were not increased by withdrawing the TGF-β/BMP signalling inhibitors Noggin and A83-01 or with the BMP4 factor in combination. However, the induced strength following the addition of the DAPT and BMP4 and removal of the Noggin and A83-01, was slightly increased compared to that of the control and weaker than that of the treatment without the BMP4 (Figure 5B,E,F). The highest proportions of ciliated cells were also induced by the DAPT plus the removal of the Noggin and A83-01, as demonstrated by flow cytometry (Appendix A, Appendix A). To test whether the ciliary function and the organoid state were affected by the in vitro induced medium, we recorded and analysed the CBF and organoid size at D8 and D16 (Figure 5C,D and Appendix A). The observation that the organoid size of the majority of the treatments at D16 was larger than that at D8 suggested that the proliferative ability of the organoids could be maintained under the inducing treatment (Appendix A). However, the combination of the DAPT and BMP4 and the removal of the Noggin and A83-01 significantly inhibited the organoid size during the induced period. On the other hand, the CBF of all the treatments was recorded at room temperature, and the average was 4 Hz. The removal of the Noggin and A83-01 slightly increased the CBF, while the DAPT did not affect the CBF. Taken together, our results further showed that the differentiation of airway stem cells into ciliated cells was regulated by Notch and TGF-β/BMP signalling. The inhibition of Notch signalling and removal of the inhibitors of TGF-β/BMP signalling could harvest the highest proportions of ciliated cells in a short time, which is highly efficient and suitable for cilia-related research. In view of the decreased immune response in the DNAH5-deficient organoids, we questioned whether the expression of cytokines could be affected by a cilia-induced treatment. Our results show that the mRNA levels of the main cytokines, G-CSF, GM-CSF, IL-6, IL-8 and TNF-α, were significantly changed in the airway organoids under the different treatments (Figure 5G). Together, we have established an optimized cilium-induced and immune-regulated condition upon the modulation of the TGF-β/BMP and Notch pathways.

## 4. Discussion

Here, we identified a PCD paediatric patient with a DNAH5 mutation and described the corresponding typical characteristics caused by cilia defects. We further constructed a patient-specific child airway organoid derived from a bronchoscopic biopsy, showing that it recapitulated the phenotype of airway epithelium with a DNAH5 deficiency in vitro. The combination of single-cell RNA-Seq, proteomic analyses and cytokine array demonstrated that a DNAH5 deficiency could impair the immune response by decreasing immune gene expression and reducing cytokine levels after a RSV infection. We further elucidated that the TGF-β/BMP and Notch pathways could induce cilia differentiation and regulate the expression of inflammatory cytokines.

The human airway is composed of a complex hollow lumen system with diverse epithelial cells lining the luminal surface, which directly interfaces with the external environment and pathogens. To maintain the airway homeostasis state, basal cells, recognized as resident stem cells, typically maintain a quiescent state. During normal airway development, or upon damage, the basal stem cells can self-renew and differentiate into other airway cell types to restore airway homeostasis. The inhaled pathogens are wrapped in the mucus, which is secreted by the club and goblet cells, and excreted by mechanical power, generated by the multiciliated cells. The concerted actions of multiple cell types constitute a first barrier to defend against the invasion of external pathogens. However, defects in this barrier always lead to frequent respiratory infections, and the typical disease is PCD. DNAH5 is an important ODA distributed along the axoneme from proximal to distal [13]. Several studies have shown that a DNAH5-deficiency always leads to immotile cilia and frequent infections in the respiratory system [9]. Apart from the cilia beating function, the inflammation and morphology of goblet cells are associated with decreased DNAH5 expression [30]. The potential mechanism and therapeutic target are still unknown. Therefore, there is an urgent need for an excellent model to recapitulate PCD with DNAH5 mutation and understand the pathophysiology.

To model the child airway epithelium with PCD, we separated the small airway tissues from the bronchoscopic biopsy of a PCD-suspected paediatric patient and constructed the child airway organoid. The striking feature of the child PCD organoid is the patient specificity, which allows the PCD organoid to be a unique experimental tool to better understand the underlying mechanisms caused by mutated genes and identify potential drug targets. With this unique model, we found that DNAH5 expression appears to closely correlate with the overall immune response of the airway epithelium. The DNAH5 deficiency significantly decreased the expression of innate and adaptive immune systems that further affected the normal release of cytokines, while only IL-8 was increased after infection with RSV. Moreover, specific cell types are also affected by DNAH5 damage, as confirmed by scRNA-Seq. For basal cells, the impaired biological processes of cell morphogenesis and cell-substrate adhesion, which are necessary for cell migration above the basement membrane and regeneration in response to injury, limit the capacity for repair [31,32]. For ciliated cells, the motile cilia length is regulated by the TGF-β pathway by affecting the transition zone of the cilium [26], while the key regulators, CTGF, ATF3, TGFBR3 and IRF7, showed low expression after damage to DNAH5, the important protein of ODA. This result indicates that ciliary function could be indirectly and directly affected by DNAH5. The secreted cytokines, regeneration of basal cells and mucociliary clearance of ciliated cells play key roles as the first defensive barrier against RSV and other pathogens. The deficiency of the first defensive barrier may be one possible reason for the frequent infections in PCD children [27,33]. Our findings were consistent with previous clinical studies in which high levels of IL-8 and increased neutrophil counts were detected in PCD patients at the time of pulmonary exacerbation compared with CF [34,35]. The increased RSV-N copies and abnormal airway inflammation could reflect an inadequate response to the RSV infection and failure to inhibit the virus in airway organoids with the DNAH5 mutation. Moreover, there is strong evidence that excessively elevated IL-8 and neutrophil counts are key elements inducing lung damage [36,37,38]. Therefore, these results provide new insights that appropriate immunological therapeutics may be another option to alleviate clinical infections in DNAH5-deficient patients.

The small number of ciliated cells in the airway organoids cultured by the standard airway organoid media greatly limited the cell functional experiments. Cilia differentiation derived from stem cells is mainly regulated by Notch, Wnt and BMP signalling [28,29,39,40]. The previous studies mostly focused on cilia-induced efficiency and not on ciliary movement, which may be affected by alterations in core signalling and viscosity changes in the closed organoid lumen, due to different proportions of mucous cells in the various treatments [22,28]. Our results demonstrated that the addition of a Notch inhibitor and removal of the TGF-β/BMP inhibitor strongly increased the ciliated cell proportion compared to the conventional airway culture media. Furthermore, the CBF was not significantly affected. This ciliated cell-induced protocol could provide more cell samples and opportunities for research on PCD and cilia-related disease.

In conclusion, we established a patient-specific child airway organoid from the biopsy of a paediatric patient with a DNAH5 mutation. This personalized model recapitulates the gene phenotype and abnormal ciliated function that provide an excellent tool to understand potential pathogenic mechanisms and explore possible therapeutic targets. We propose child PCD airway organoids as a valuable and efficient preclinical and personal model for academic, clinical, pathophysiological and pharmaceutical applications.

## Figures and Tables

**Figure 1 cells-11-04013-f001:**
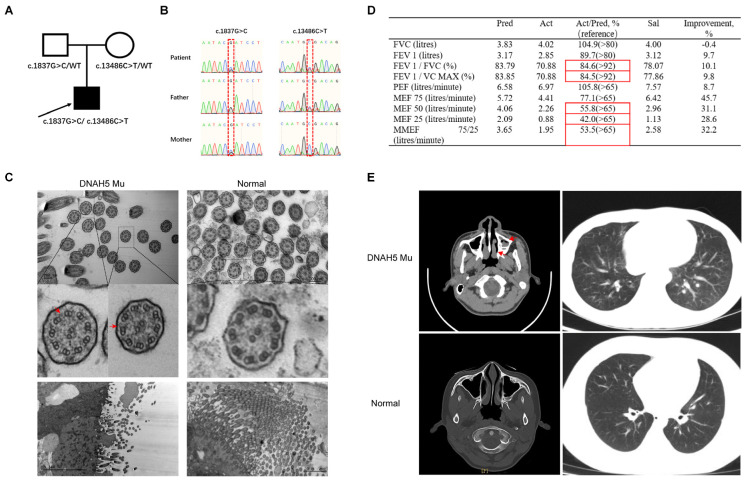
The characteristics of a paediatric patient with a DNAH5 mutation. (**A**) The pedigree structure of the family with DNAH5 mutation. The black squares represent the proband. (**B**) Sanger sequencing identifying the compound mutations in the patient and mutation-carrier parents. (**C**) Ultrastructure of the ciliary axonemes from a healthy child and the patient by TEM showing the “9 + 2” structure. The red arrow reflects the partial defect of the outer dynein arm. (**D**) The results of spirometry in the patient with PCD demonstrating moderate obstructive ventilation dysfunction and small airway airflow obstruction. Pred: predicted value; Act: actual measured value; Act/Pred: ratio of actual measured to predicted values; Sal: measured value after inhaling salbutamol; Improvement: ratio of Sal/Pred to Act/Pred; FVC: forced vital capacity; FEV1: forced expiratory volume in one second; FEV1/FVC: ratio of FEV1 to FVC; FEV1/VC MAX: ratio of FEV1 to VC MAX; PEF: peak expiratory flow rate; MEF 75: maximal expiratory flow after 75% of the FVC has not been exhaled; MEF 50: maximal expiratory flow after 50% of the FVC has not been exhaled; MEF 25: maximal expiratory flow after 25% of the FVC has not been exhaled; MMEF 75/25: maximal mid-expiratory flow. The measurements of FEV1/FVC and FEV1/VC MAX were 84.6% and 84.5%, respectively, suggesting a moderate obstructive ventilation dysfunction. The MEFs of 50 and 25 were less than 65%, suggesting a small airway airflow obstruction. The reference range is shown in brackets; the red boxes indicate abnormal values. (**E**) The nasopharyngeal CT scan of the nasal sinuses shows pansinusitis in the patient. The red arrow reflects nasosinusitis.

**Figure 2 cells-11-04013-f002:**
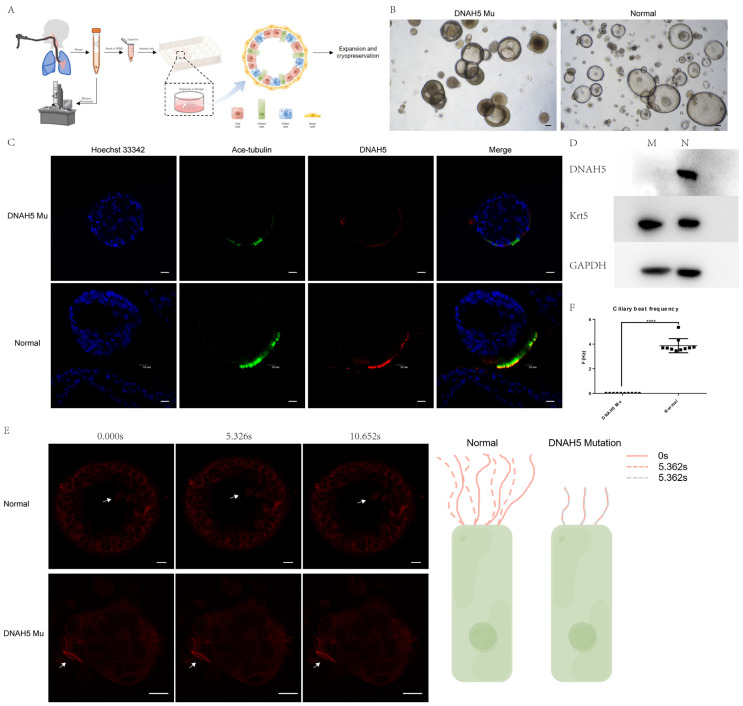
Patient-specific child airway organoids from a paediatric patient with a DNAH5 mutation. (**A**) Schematic workflow of child airway organoid generation derived from bronchoscopic biopsy. (**B**) Bright-field images depicting child airway organoid phenotypes. Scale bar = 100 µm. (**C**) Whole-mount child airway organoid immunofluorescence shows DNAH5 expression on the cilia. Hoechst 33342, blue; DNAH5, red; acetylated tubulin, green. Scale bar = 10 µm. (**D**) Western blot analysis showing the protein expression of DNAH5 in airway organoids from PCD patients and healthy controls. Krt5 is a marker protein of basal cells; GAPDH is the endogenous control. (**E**) SPY555-tubulin live imaging showing the difference in cilia motility between normal and PCD airway organoids. Normal cilia are motile (**upper**), whereas the cilia in PCD airway organoids with DNAH5 mutation are immotile (**lower**). The right part is a schematic representation. The white arrows reflect the cilia, and the time was measured in seconds(s). Scale bar = 10 µm. (**F**) Ciliary beat frequency of normal and PCD airway organoids with DNAH5 mutation. The cilia in PCD airway organoids are immotile. **** *p* < 0.0001. n = 10 per group.

**Figure 3 cells-11-04013-f003:**
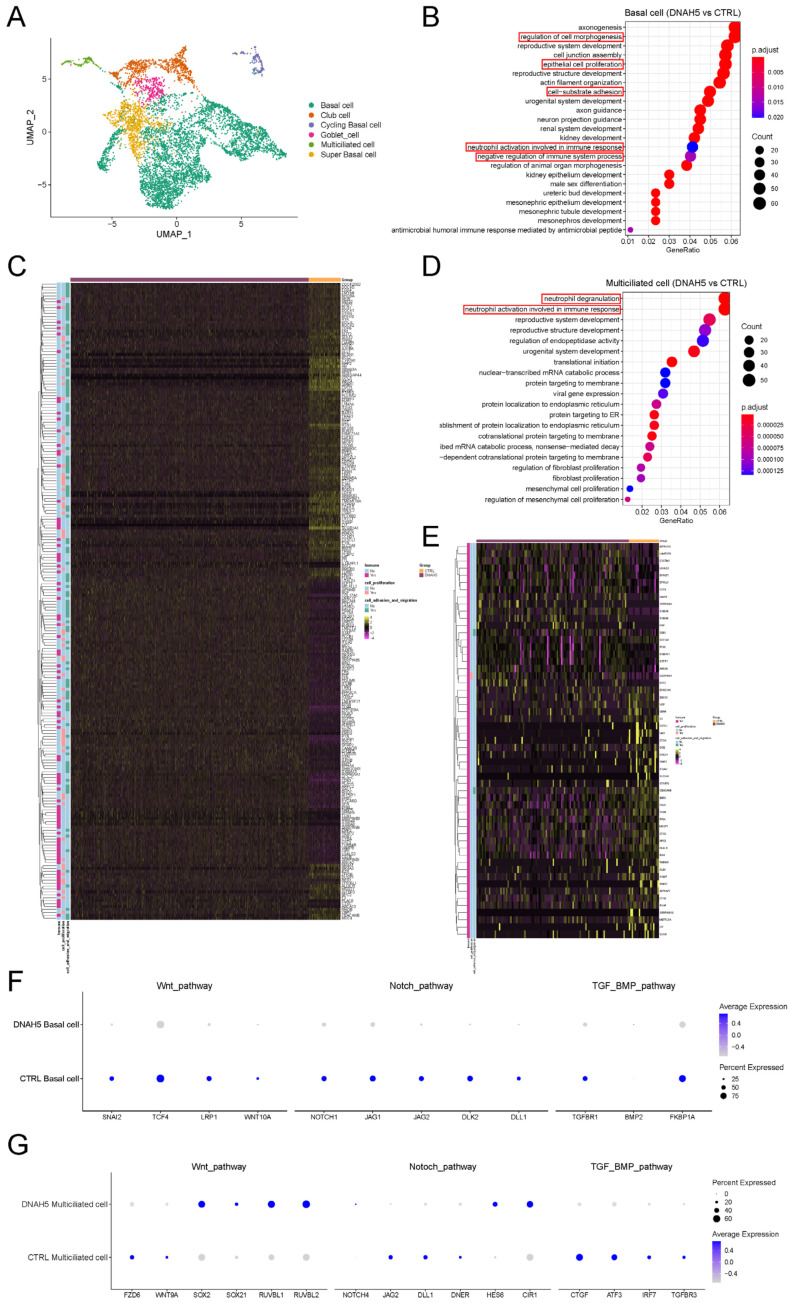
Single-cell transcriptional profiles from child PCD organoids. (**A**) UMAP plots of expression from scRNA-Seq of child airway organoids (DNAH5-mutated and control). (**B**) GO enrichment analysis of biological processes in basal cells (DNAH5 vs. CTRL). (**C**) Heatmap of the differentially expressed genes in immune response, regulation of cell morphogenesis, cell-substrate adhesion and epithelial cell proliferation between DNAH5 mutation and normal organoids. (**D**) GO enrichment analysis of biological processes in multiciliated cells (DNAH5 vs. CTRL). (**E**) Heatmap of the differentially expressed genes in the immune response between DNAH5-mutated and normal organoids. (**F**,**G**) Dot plot of Wnt, TGF-β/BMP and Notch pathway-regulated genes in basal cells and multiciliated cells in DNAH5-mutated and normal airway organoids. The colour gradient (grey to blue) and dot size indicate the mean marker expression and the percentage of cells expressing the marker for each cluster.

**Figure 4 cells-11-04013-f004:**
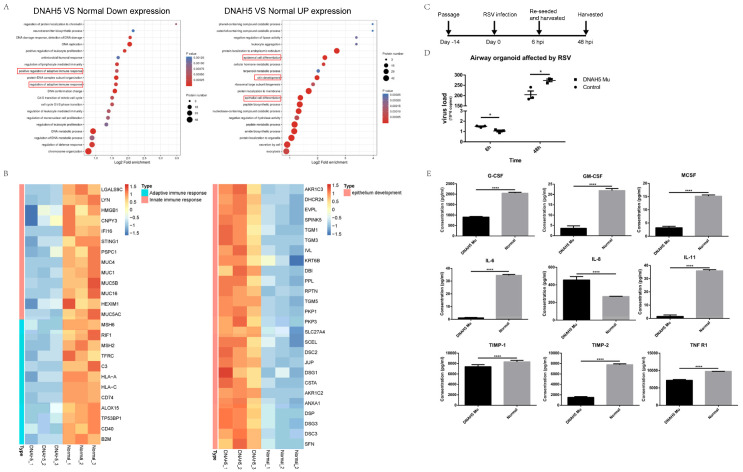
Decreased immune response in DNAH5-mutated organoids. (**A**) GO enrichment analysis of biological processes in DNAH5-mutated and normal airway organoids. The immune responses were downregulated in DNAH5-mutated airway organoids (**left**), whereas epithelial cell differentiation was downregulated (**right**). (**B**) The heatmap illustrates the expression level changes of the statistically significant proteins related to the adaptive immune response, innate immune response and epithelial development between DNAH5-mutated and normal airway organoids. (**C**) Stepwise protocol for airway organoids infected by RSV. (**D**) The copies of RSV-N detected by ddPCR confirming the different infections between DNAH5-mutated and normal airway organoids. * *p* < 0.05; n = 3 replicates. (**E**) The expression of immune factors in DNAH5-mutated and normal airway organoids infected by RSV at 48 hpi. **** *p* < 0.0001; n = 3 replicates.

**Figure 5 cells-11-04013-f005:**
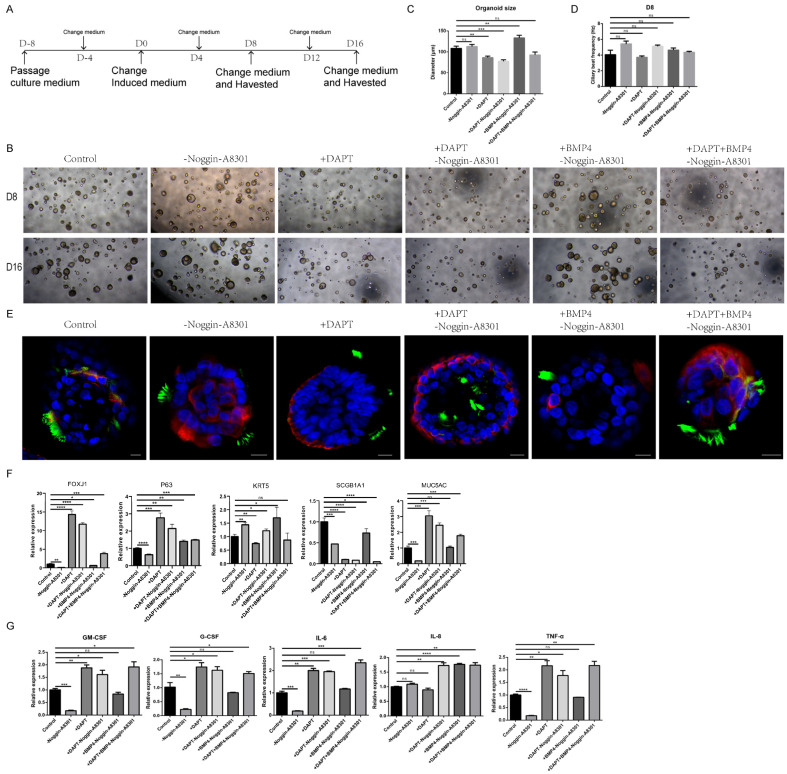
Cilia-induced strategies regulating cell differentiation in airway organoids. (**A**) Stepwise protocol for cilia differentiation in normal airway organoids. (**B**) Bright images of normal child airway organoids after different cilia-inducing treatments for 8 days (D8) and 16 days (D16). (**C**) The organoid size in bright images (**B**) was measured at D8. ** *p* < 0.01, *** *p* < 0.001, ns: no significance; n = 20 per group. (**D**) The ciliary beat frequency of normal child airway organoids in bright images (**B**) was recorded at D8. ns: no significance; n = 20 per group. (**E**) Whole-mount immunofluorescence organoids show ace-tubulin protein expression in the normal child airway organoid after different cilia-inducing treatments for 8 days. Hoechst33342, blue; krt5, red; acetylated tubulin, green. Scale bar = 10 µm. (**F**) Relative mRNA expression of marker genes in normal child airway organoids after different cilia-inducing treatments for 8 days. Ciliated cell (FOXJ1), basal cell (KRT5, P63), club cell (SCGB1A1), goblet cell (MUC5AC). * *p* < 0.05, ** *p* < 0.01, *** *p* < 0.001, **** *p* < 0.0001; ns: no significance; n = 3 replicates. (**G**) The mRNA relative expression of immune cytokines in normal child airway organoids after different cilia-inducing treatments for 8 days. * *p* < 0.05, ** *p* < 0.01, *** *p* < 0.001, **** *p* < 0.0001; ns: no significance; n = 3 replicates.

## Data Availability

The mass spectrometry proteomics data have been deposited with the ProteomeXchange Consortium via the PRIDE partner repository with the dataset identifier PXD038024. The single-cell RNA sequencing data reported in this paper have been deposited in the NCBI GEO database under the accession number GSE217596.

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
