# Peer review of "Multiomics Analysis of a DNAH5-Mutated PCD Organoid Model Revealed the Key Role of the TGF-β/BMP and Notch Pathways in Epithelial Differentiation and the Immune Response in DNAH5-Mutated Patients"

_cells, 2022, doi:10.3390/cells11244013_

Round 1

Reviewer 1 Report

In the manuscript entitled "Multiomics analysis of a DNAH5-mutated PCD organoid model revealed the key role of the TGF-β/BMP and Notch pathways in epithelial differentiation and the immune response in DNAH5-mutated patients",  a PCD airway organoid was estabilished from the bronchoscopic biopsy of a patient with DNAH5 mutation, and the effects of DNAH5 mutation were investigated using scRNA-seq and proteomic analysis. Further, TGF-β/BMP and Notch pathways were found to be involved in the epithelial differentiation and the immune response in DNAH5-mutated patients. The personalized organoid model is informative for exploring both pathological mechanisms and intervention strategies.

The manuscript is well organized and further processing could be carried out after minor revisions.

- The full name should be provided for the abbravation when first ammentined, such as ddPCR.

- Section 2.16, the method used in multiple comparation following one-way ANOVA should be provided.

Reviewer 2 Report

The manuscript investigates how a mutation in DHAH5 affects the immune profile in PCD airway cells using DNAH5-mutated airway organoids derived from a PCD patient. The paper did a good job establishing the working model and characterization of the cells.  There are several questions that need to be addressed.

1 The label for each panel in figure1 is not consistent with the figure legend and the text.

2 The TEM data does not provide the resolution to look at the outer dynein arm. To support the statement in the result that there is a partial defect of the outer dynein arm, please show zoom-in data for the mutant group and control. 

3 In figure2C, Ace-tubulin staining showed that the cilia numbers were significantly reduced in the patient sample with DNAH5 mutation. However, the cell number in the mutant group is much less than in the control group. How does the author rule out the possibility that reduced cilia number is due to reduced cell number in the mutant group? How many replicates are there for the experiment in figure 2C and figure 2D?  Please show the quantification data with individual data points in the bar graph.

4 The data in figure 2E is not clear and confusing. It is hard for readers to get any useful information from this figure. 

5 The text in figure3 is too small to read. 

6 RSV-N gene copies in airway organoids with DNAH5 mutation were significantly lower than those in the control at the initial infection stage. What is the potential reason for this result? How reduced cilia affect virus infection? This is an interesting discussion point. 
